# Glycation of Whey Proteins Increases the Ex Vivo Immune Response of Lymphocytes Sensitized to β-Lactoglobulin

**DOI:** 10.3390/nu15143110

**Published:** 2023-07-12

**Authors:** Dagmara Złotkowska, Mateusz Kuczyński, Ewa Fuc, Joanna Fotschki, Barbara Wróblewska

**Affiliations:** Institute of Animal Reproduction and Food Research Polish Academy of Sciences in Olsztyn, Department of Immunology and Food Microbiology, Tuwima 10, 10-748 Olsztyn, Polande.fuc@pan.olsztyn.pl (E.F.); j.fotschki@pan.olsztyn.pl (J.F.); b.wroblewska@pan.olsztyn.pl (B.W.)

**Keywords:** glycation, bovine β-lactoglobulin, α-lactalbumin, T cells, mice C57BL/6

## Abstract

Glycation is a spontaneous reaction accompanying the thermal processing and storage of food. It can lead to changes in the allergenic and immunogenic potential of protein. This study aimed to evaluate the effect of the glycation of α-lactalbumin and β-lactoglobulin (β-lg) on the ex vivo response of β-lg sensitized lymphocytes. C57BL/6 mice were immunized intragastrically (i-g) or intraperitoneally (i-p) with β-lg. The humoral response of the groups differed only with respect to the IgE level of the i-p group. Cellular response was studied after stimulation with antigen variants. The lymphocytes from the i-g/group mesenteric lymph nodes, stimulated with β-lg before and after glycation, presented a higher percentage of CD4 and CD8 T cells compared to the i-p/group. The cytokine profile of the i-p/group splenocytes stimulated with antigens showed elevated levels of pro-inflammatory IL-17A regardless of protein modification. In conclusion, the ex vivo model proved that the glycation process does not reduce protein immunogenicity.

## 1. Introduction

Cow’s milk proteins consist of about 82% casein (CN) and about 12% whey proteins (β-lactoglobulin (β-la) and α-lactalbumin (α-la)). Cow’s milk can be used as a substitute for human milk in the diets of newborns and babies due to its nutritional value. Cow’s milk proteins are usually the first foreign proteins entering a newborn’s gut and sometimes trigger an overreaction of the immune system. According to the WHO’s allergens list, milk proteins are strong allergens, with caseins and β-lg being the most allergenic. CN has shown stronger allergenic potential than β-lg, triggering a significantly higher IgE response [1,2]. Cow’s milk allergy (CMA) an ailment that is difficult to diagnose because it is associated with various symptoms like emesis, diarrhea, reflux, urticaria, hives, and eczema. When left untreated, CMA can lead to enteropathy or proctocolitis, and severe cases can cause life-threatening anaphylaxis. To date, an elimination diet is the only reliable method of preventing allergic reactions to cow’s milk. We came across a report describing a study on mice with induced cow’s milk allergy; the study revealed the possibility of lowering the levels of two allergy markers, class E antibodies and interleukin 4, via the long-term administration of horse milk to sensitized mice [3].

Milk-processing methods include thermal processes and the application of high pressure, which lead to the degradation or aggregation of cow’s milk proteins’ structures and mask or expose their structural epitopes [4,5]. Milkovska-Stamenova et al. [6] claimed that storage and heating affect the intensity of glycation with respect to milk proteins. Glycation (called the non-enzymatic glycosylation or the Maillard reaction) is a spontaneous chemical reaction between a reducing monosaccharide and the primary free amino group of the terminal amino acid in a protein. In the final stage, advanced glycation end-products (AGE) are created [7] [Figure 1].

Under homeostatic conditions, AGEs are found in low concentrations, and the body removes them via degradation in proteasomes. Raubach et al. [8] demonstrated that the glycation conditions of myoglobin (such as varying glucose concentrations) affected its proteasomal degradation. The accumulation of AGE in tissues is associated with a number of processes (e.g., glycation increases the stiffness of collagen fibers), the development of cardiovascular diseases (high levels of blood glucose increase the frequency with which blood glucose binds to proteins such as hemoglobin or low-density lipoprotein (LDL)), the development of metabolic disorders, diabetes and Alzheimer’s disease (via the degradation of some brain proteins), and kidney diseases [8,9].

Glycation affects the functional and biological properties of food proteins. Scientific evidence has shown that the glycation of milk proteins effectively changes their structures and immunogenicity. One might expect that glycation could be a promising and safe method for masking food protein allergenicity [10,11]. Aoki et al. (2006) [12] observed conformational changes upon conjugation with the cationic saccharide chitosan, effectively reducing the immunogenicity of β-lg. Zhang et al. [10] suggested that the reduction in the antigenicity of glycated α-la is higher with the increase in the size of the saccharide. However, protein glycation can also generate new epitopes (neo-allergens) or change existing ones recognized by IgE and IgG [13]. The IgE-binding ability of allergens modified by glycation was studied; however, the results were equivocal. Taheri-Kafrani et al. [14] showed that the moderate glycation of β-lg in the early stages of the Maillard reaction slightly affects its recognition by IgE. In contrast, in that study, a high degree of glycation had an apparent “masking” effect on the recognition of epitopes.

The spontaneous glycation process appears to be a good method of reducing the antigenicity of milk proteins. Xu et al. [15] studied WPI glycation with dextran of different molecular weights. The authors observed reductions in the binding of specific IgE antibodies to whey protein isolate after WPI-DX glycation. Other reports have shown the possibility of reducing the antigenicity of whey proteins by implementing combined technologies; for example, glycation with phosphorylation altered the antibody response to α-la [16], and the ultrasonic pretreatment of α-la combined with dry glycation reduced IgE- and IgG-binding capacity [17] and hyposensitization of β-lg [18]. Deng et al. [19] demonstrated that glycation influences the affinity of allergens for specific IgE antibodies, which, in turn, impacts the electric charge and hydrophobicity of proteins. The study by Li et al. [20] revealed that the glycation process decreased the IgE/IgG-binding capacity of α-la and, when combined with dynamic high-pressure microfluidization, was a promising solution for reducing the allergenicity of proteins.

The mucosal immune system is the first point of contact with digested antigens. In the lymphoid inductive sites (mesenteric lymph nodes—MLN; Peyer’s patches—PP), antigens are recognized, and an immune response is induced. The process involves mast cells, dendritic cells, and B and T cells. Signaling molecules, like cytokines, are released to profile T cell populations, i.e., CD4, CD8, CD25, and regulatory T cells CD4^+^CD25^+^FoxP3^+^. Depending on an antigen’s properties, immunity, tolerance, or allergy may be induced [21].

To date, there is no clear evidence with which to predict the influence of AGE derived from food on the allergenicity of proteins. Detailed studies must still be conducted in order to determine the immunological characteristics of glycated proteins derived from cow’s milk. Therefore, this study aimed to evaluate the in vitro effect of glycated β-lg and α-la on the immunocompetent cells of C57BL/6 mice immunized with β-lg. The intraperitoneal administration of allergens in laboratory animals has been employed for many decades, and it is a common practice in many in vivo studies of disease models, especially with respect to allergies. In our study, we tested the immune response of β-lg-sensitized lymphocytes (intragastric or intraperitoneal) to modified whey proteins. We investigated differences in the mechanisms of immune regulation induced by whey proteins. An ex vivo study of changes in T cells profile, lymphocyte proliferation, and the profile of released cytokines in response to glycated (possible neo-allergens) and non-glycated β-lg and α-la will contribute to the knowledge regarding the functional changes and safety of modified proteins and food containing cow’s milk proteins.

## 2. Materials and Methods

### 2.1. Materials

The α-la and β-lg used in this study were purchased from Sigma-Aldrich (St. Louis, MO, USA). All antigens were of analytical grade. Glucose (49163, Sigma-Aldrich) and lactose (17814, Sigma-Aldrich) were used for the glycation process.

### 2.2. Glycation Protocol and Product Characterization

α-la and β-lg solutions containing glucose or lactose (for which the molar ratio of protein to sugar was 1:10 (*w*:*w*)) were sterilized via filtration and incubated at 50 °C for 35 days (α-la/G; α-la/L; β-lg/G; β-lg/L). Thermally treated samples without saccharides (α-la/H; and β-lg/H) served as controls for the process. After thermal treatment, the samples were transferred to dialysis tubes (SEPKO, MWCO 8000) and dialyzed against water for 72 h at 4 °C. The water was changed every 24 h. Then, the samples were lyophilized and stored at 4 °C until further analysis.

The degree of glycation was calculated according to the amount of sugar bound to protein molecules estimated via the periodate method described by Ahmed and Furth [22]. Briefly, 40 μL of sample was incubated with 20 μL 0.1 M HCl and 20 μL 0.05 M NaIO_4_ for 30 min at room temperature. The oxidation process was stopped by placing the samples on ice for 10 min; then, 20 µL of chilled 15% ZnSO_4_ solution and 20 µL of 0.7 M NaOH solution were added. Samples were centrifuged for 10 min at 9000× *g* at room temperature. A total of 100 µL of supernatant was added to 200 µL of freshly prepared reagent. After incubation for 1 h at 37 °C in the dark, the absorbance was read at 405 nm using a Jupiter UVM−340 spectrophotometer (ASSYS Hitech, GmbH, Eugendorf, Austria). The bound sugar was calculated from a standard curve prepared analogously (sugar concentration range 0–100 nM/100 µL).

#### SDS-Polyacrylamide Gel Electrophoresis (SDS-PAGE)

We employed the standard SDS-PAGE protocol described elsewhere [7]. Briefly, protein samples (up to 20 μg of protein) and Precision Plus Protein Standards™ (1610363, BIO-RAD) were loaded on 15% gel and separated under a constant current of 24 mA using the Mini-PROTEAN system (Bio-Rad Laboratories, Inc., Warsaw, Poland). The gel was stained with 0.1% Coomassie Brilliant Blue R-250 (Sigma-Aldrich). After being de-stained, the gels were scanned using a Pharos FX Plus Imager and analyzed using Quantity One software v. 4.6.1 (Bio-Rad, Hercules, CA, USA).

### 2.3. In Vivo Experiment

Six-week-old female C57BL/6 inbred mice were purchased from the Centre of Experimental Medicine in Białystok (Poland). They were maintained in the animal care facility at the Institute of Animal Reproduction and Food Research in Olsztyn (Poland). Mice were fed a maintenance diet designed for rats and mice (#1320 TPF, Altromin, Lage, Germany) that was free from milk proteins. Water and food were provided ad libitum. Animals were randomly assigned into two groups (*n* = 8/group). Mice were immunized intraperitoneally (i-p) with 200 μg of β-lg containing Freund’s adjuvant and intragastrically (i-g) with 20 mg of β-lg containing cholera toxin (CT) as a mucosal adjuvant. On day 26, mice were terminated via CO_2_ inhalation in a chamber with a flow rate of 10% of the chamber volume per minute [1]. Fecal, blood, and tissue samples were collected from individual mice according to standard protocols [7].

#### 2.3.1. Serum and Fecal Sample Preparation

After coagulation, blood samples were centrifuged at 16,900× *g* and 10 °C for 10 min using an Eppendorf 5418R device (Eppendorf, Hamburg, Germany).

Fresh fecal pellets were suspended in 0.1% NaN_3_ in PBS at a ratio of 100:1 (*w/v*); then, they were extracted via mixing at 4 °C for 10 min (Fugamix^®^, ELMI ILD, Latvia) and centrifuged (16,900× *g*, 10 °C, 10 min). Supernatants were collected and stored at −20 °C for subsequent analysis.

#### 2.3.2. Lymphocyte Isolation

For the processing of the lymphocytes, we used two kinds of media:Complete Media (CM)—RPMI-1640 (G6784, Sigma-Aldrich) supplemented with 10% heat-inactivated fetal bovine serum (FBS), 1 mM of nonessential amino acids, 1 mM of HEPES, 1 mM of sodium pyruvate, and 10 units/mL of penicillin–streptomycin solution.Incomplete Media (IM)—RPMI-1640 supplemented with 1 mM of HEPES, 1 mM of sodium pyruvate, and 10 units/mL of penicillin–streptomycin solution.

All chemicals used for lymphocyte preparation and treatment were purchased from Sigma-Aldrich and were cell-culture grade.

#### 2.3.3. Isolation of Peripheral Blood Mononuclear Cells (PMBC)

Terminal blood, extracted via heart puncture, was transferred to a test tube and mixed with 20 µL of Heparinum WZF (5000 IU/mL, Polfa, Warsaw, Poland) to prevent clotting. The samples were diluted with PBS 1:1 (*v*/*v*) and separated via density gradient using Histopaque^®^-1077 (Sigma-Aldrich). The interface layer containing the mononuclear cells was isolated and washed in IM, after which the pellet was resuspended in 1 mL of IM to allow us to count the number of cells.

#### 2.3.4. Isolation of Lymphocytes from Tissues

Splenocytes (SPL), mesenteric lymph nodes (MLN), Peyer’s patches (PP), and head and neck lymph nodes (HNLN) were isolated from mice. Tissues were placed in IM solution, homogenized in a glass tissue douncer, and filtered through an 80 µm nylon filter to remove cell debris. Additionally, splenocytes were incubated for 5 min with red blood cell lysis buffer (Sigma-Aldrich) to remove erythrocytes. Then, cells were washed and centrifuged at 413× *g* (5418R, Eppendorf, Hamburg, Germany) at 10 °C for 10 min and suspended in 1 mL of IM [7]. Cell suspensions were stained with 0.4% Trypan Blue solution (Sigma-Aldrich). Then, the Bürker Cell Counter was used to count the number of lymphocytes.

### 2.4. Lymphocyte Culture

Lymphocytes were plated on 96-well microplates at a concentration of 1 × 10^6^ cells/100 μL in CM. After 12 h of culturing at 37 °C and 5% CO_2_, cells were stimulated with antigens (α-la, α-la/G, α-la/L, β-lg, β-lg/G, and β-lg/L) at a concentration of 200 µg/mL. Concanavalin A (ConA), at a concentration of 10 µg/mL, and cells growing in a medium served as controls. After 120 h, cells were collected and centrifuged at 413× *g*/10 °C for 10 min (Eppendorf). The supernatant was collected and stored at −80 °C for cytokine assay. Cells were subject to lymphocyte phenotyping (p.2.6).

### 2.5. Lymphocyte Proliferation Assay

The lymphocyte proliferation index (PI) was determined using the flow cytometry CFSE method. Briefly, the carboxyfluorescein diacetate N-succinimidyl ester (CFSE, Sigma-Aldrich) was dissolved in dimethyl sulfoxide to obtain a stock solution of 5 mM; then, it was stored at −20 °C. For dye labelling, lymphocytes were resuspended in 1 mL of RPMI-1640 medium with 5% heat-inactivated FBS and carefully transferred to fresh tubes. Subsequently, 1.1 µL of the CFSE stock solution was added to the cells and incubated for 5 min at 20 °C in the dark. Cells were washed twice, once in PBS supplemented with 5% FBS and once in PBS supplemented with 1% FBS, and then washed again. Then, the cells were seeded on a 96-well microplate at a concentration of 1 × 10^6^ cells/100 μL in CM and stimulated with 200 µg/mL of antigen. After 120 h of incubation at 37 °C and 5% CO_2_, lymphocytes were additionally washed and stained with rat anti-mouse CD4 antigen (PerCP-Cy 5.5, Clone RM4-5, BD Pharmingen, San Diego, CA, USA) and propidium iodide (BD Pharmingen). The samples stimulated with 10 μg/mL ConA served as a positive control for the assay. Fifty thousand events were collected from each sample using a BD LSR Fortessa Cell Analyzer (BD Biosciences, San Jose, CA, USA) equipped with DIVA software (BD Bioscience, San Jose, CA, USA). Results were analyzed using FlowJo software (FlowJo LLC, Ashland, OR, USA) with a proliferation platform.

### 2.6. Lymphocyte Phenotyping

Cellular and intracellular markers used in the presented study were produced by BD Pharmingen^®^ (San Diego, CA, USA). We used a freshly prepared cocktail of antibodies, namely, FITC anti-mouse CD4 (553929, clone H129.19), PerCP-Cy 5.5 anti-mouse CD25 (551071, clone PC61) and AF700 anti-mouse CD8a (557959, clone 53-6.7), for standard lymphocyte staining. Briefly, cells were incubated at 4 °C for 15 min and washed with FACS buffer (PBS supplemented with 5% FBS). For intracellular marker Foxp3 staining, cells were fixed in 2% paraformaldehyde, washed with FACS buffer, permeabilized using ice-cold methanol for 20 min at room temperature, rewashed, and stained with AF-647 anti-mouse Foxp3 (560401, clone MF23). After each step, samples were centrifuged at 413× *g*/10 °C/10 min (5418R, Eppendorf, Hamburg, Germany). The samples were analyzed using a BD LSR Fortessa Cell Analyzer (BD Biosciences, San Jose, CA, USA) equipped with DIVA software (BD Bioscience, San Jose, CA, USA). The data were analyzed using FlowJo 10.3 (FlowJo, LLC) software. Fifty thousand events were collected from each sample. The gating tree was set as follows: FSC/SSC (forward and side scatter, representing cell distribution by size and intracellular composition) lymphocytes were gated in the range of 100 to 150 kDa; then, CD4^+^ and CD8^+^ were selected. The CD4^+^ T cells were additionally gated for CD25^+^, and double-positive CD4^+^CD25^+^ was gated for Foxp3^+^. Results are presented as the means of the groups ± SD.

### 2.7. Determination of Specific Immunoglobulin Classes G and A in Serum Samples and Fecal Extracts

Standard indirect ELISA was used to determine the titers of specific immunoglobulins (IgG, IgG1, IgG2b, IgG3, and IgA). Briefly, 96-well plates were coated with β-lg antigen (20 μg/mL) and incubated at 37 °C for 1.5 h. Then, the remaining free sites were blocked with 1% BSA (bovine serum albumin) in PBS, incubated at 37 °C for 1 h, and washed 3 times in PBST (PBS + 0.5% Tween 20). After washing, serum samples or fecal extracts in suitable solutions were added to the plate and incubated overnight at 8 °C. After the washing procedure, horseradish-peroxidase (HRP)-labeled anti-mouse IgG, A (Sigma-Aldrich) was added to each well and incubated for 1 h. Finally, we used substrate ABTS (Millipore, Temecula, CA, USA) to visualize the reactions. After 1 h of incubation, absorbance was measured at 405 nm using a Jupiter UVM−340 spectrophotometer (ASSYS Hitech, GmbH, Eugendorf, Austria). Endpoint titer (EPT) values were expressed as the reciprocal dilution of the last sample dilution of 0.1 OD (optical density) above the negative control [1].

### 2.8. Cytokine Profile

The concentrations of the cytokines IL-2, IL-4, IL-6, IL-10, interferon-gamma (IFN-γ), tumor necrosis factor (TNF-α), and IL-17 released to the culture medium were evaluated using a BD Cytometric Bead Array Mouse Th1/Th2/Th17 Cytokine Kit (560485, BD Biosciences) and analyzed on a BD LSR Fortessa Cell Analyzer. Data were analyzed using FCAP Array 3.0 software (BD Biosciences).

### 2.9. Statistical Analysis

Statistical analysis was performed using GraphPad Prism version 8.0.0 (GraphPad Software, San Diego, CA, USA). Results are presented as means of the groups ± SD. For statistical analysis, tests were applied according to the number of experimental groups compared. For the comparison of two groups, T-test was used, while one-way or two-way ANOVA followed by post hoc Tukey’s test were applied for multiple comparisons. The differences were considered statistically significant at *p* ≤ 0.05.

## 3. Results and Discussion

The research presented herein describes how whey protein glycation affects the cellular immune response. The strength of this research is its target, that is, the glycation process of milk proteins. Glycation is a spontaneous process that accompanies the thermal processes carried out the food industry. Milk whey proteins (α-la and β-lg) are a nutrient-rich raw material commonly used in the food industry. Our study used a model system of single proteins, which limited the possibility of assessing the influence of interactions between proteins on the progress of glycation in addition to the competitiveness of proteins in reaction with sugar. Glycation in the model system reduced the number of variables employed in the experiment that affect the induction of immunocompetent cells. This simplification of the experiment allowed us to determine the trends of changes in the immune system response at the cellular level.

### 3.1. Glycated Proteins

The SDS-PAGE analysis and an analysis of the amount of sugar bonded to proteins were performed on samples heated with and without sugar (heated and glycated) (Figure 1A,B). The protein bands of the thermally treated (α-la/H) and glycated (α-la/G and α-la/L) α-la samples showed a migration rate corresponding to a molecular weight of 14 kDa (Figure 1A).

The β-lg samples’ (β-lg/H; β-lg/G; β-lg/L) pattern includes bands with MW in the 15–18 kDa range. The bands with a higher MW (~25 kDa for α-la and around 36 kDa for the β-lg samples) correspond to protein dimers. The bands with a molecular weight >50 kDa correspond to aggregates formed during long protein heat treatment [23,24]. The broadening of the protein bands of the glycated samples (α-la/G and α-la/L; β-lg/G; and β-lg/L) revealed changes followed by sugar molecule binding [25].

To establish the progress of glycation, we determined the amount of sugar bound to a molecule of each protein (Figure 1B). We found more glucose bound to α-la after glycation than lactose (1.78 ± 0.02 vs. 1.56 ± 0.003 mol/mg (*p* < 0.05)). Under the same conditions, β-lg binds more sugar, which proves that it undergoes glycation more readily than α-la, independent of the kind of sugar. This is in line with the findings of Gazi et al. [25], who reported that the compact structure of α-la makes it more resistant to processes like denaturation, proteolysis, and glycation. Ledesma-Osuna et al. [26] studied BSA glycation in glucose, galactose, and lactose and demonstrated that galactose and glucose preferentially bind to BSA over lactose. To summarize, our results indicate that glycation occurred in all the samples heated with sugar.

### 3.2. Humoral Response against β–lg

The results showed that β-lg immunization induced similar levels of specific IgG (in serum) and IgA (in serum and feces) independently of the form of antigen administration (anti-𝛽-lg EPT were IgG 29 ± 0.71 and 29.6 ± 1.34 and IgA 23.8 ± 0.45 and 23.6 ± 0.54 for the i-g and i-p routes, respectively (Figure 2A)).

The i-p immunization of mice with β-lg induced a higher amount of IgE, 29.06 ± 0.47 ng/mL, than that in the mice from the i-g group, 8.34 ± 0.96 ng/mL (*p* < 0.0001). At that exact moment, we observed a significantly higher value of specific IgG1, namely, 212.8 ± 1.30 (*p* < 0.05), after i-p treatment. Higher titers of specific IgG2a and IgG3, at 25.44 ± 1.42 and 27.78 ± 0.97, respectively, were noted in the sera of mice immunized intragastrically. The IgG profile corresponds to a protein’s affinity to FcγR receptors and the possibility of anaphylaxis shock following contact with food antigens [27]. Oral exposure to antigens involves barrier tissue immune cells such as mast cells or basophils, which play a crucial role in host defense in Th2 response and inflammatory reactions [26]. The i-p immunization triggered a significantly higher level of IgE secretion (almost three-fold) than the i-g treatment, which suggests a more severe reaction. This more robust and consistent IgE response than the intragastric administration of milk proteins has been previously observed by Cárdenas-Torres et al. [28]. However, the intragastric administration of β-lg induced high amounts of IgG2a and IgG3, suggesting the induction of a significant (*p* < 0.001) mucosal immune response to dietary proteins [29].

### 3.3. The Route of Antigen Exposure Modulates the Cellular Immune Response

The administration of antigens induces antigen-specific cellular and mucosal responses. The profiles of T cells (e.g., CD4 and CD8) in inductive (MLN and PP) and effector (SPL) lymphoid tissues are crucial to this process. They reflect the response to a given molecule with respect to immunity, allergy, or tolerance [29].

Figure 3A,B show the profiles of CD4^+^ and CD8^+^ T lymphocytes in the tissues from the mice i-p and i-g immunized with β-lg. CD4^+^ T cells are directly involved in developing the immune response to antigens.

In a presented experiment, we observed a significantly higher percentage of CD4^+^ T cells in the i-p group compared to the i-g (*p* > 0.05) group in the examined tissues (Figure 3A). For both groups, the highest levels were found in MLNs, presenting levels equal to 21.35 ± 0.15% and 28.4 ± 0.5% for the i-g and i-p groups, respectively (*p* < 0.0001). The highest levels of CD8^+^ T cells were also found in MLNs, namely, 23.4 ± 1.5% for the i-g group and 14.35 ± 3.05 for the i-p (*p* < 0.05) group, while the same values in PP were 1.97 ± 0.14% and 14.98 ± 6.4% for the i-g and i-p groups, respectively (Figure 3B). This result confirms that MLNs are the induction site of the immune response to β-lg. Additionally, we found that oral β-lg exposure increased the CD4^+^CD25^+^ population in MLNs by about eight times (amounting to 5.72 ± 0.2%) compared to the i-p group 0.65 ± 0.03% (*p* < 0.0001) (Figure 3C). In peripheral lymph nodes, i.e., the HNLN, the content of CD4^+^CD25^+^ increased almost 16-fold, amounting to 3.98 ± 0.05% in the i-g group compared to 0.240 ± 0.001% in the i-p group (*p* < 0.0001). The presence of the FoxP3 marker on CD4^+^CD25^+^ cells renders the cells critical for preventing excessive immune activation and autoimmune response initiated by reactive T cells [30]. The CD4^+^CD25^+^FoxP3 T cell population indicates the emergence of regulatory lymphocytes (Treg), suggesting an enhanced immune response to the circulating allergen 𝛽-lg. Oral allergen delivery increased Tregs induction in MLN (25.75 ± 0.35%) and peripheral tissue in addition to HNLN (18.0 ± 0.1%) compared to ~1.4% in MLN and ~4% in HNLN after i-p 𝛽-lg exposure. The significant percentage of the pivotal Treg population that was also found in PMBC (*p* < 0.0001; Figure 3D) proves allergic inflammation occurs after oral 𝛽-lg delivery.

Oral exposure to 𝛽-lg resulted in the functional differentiation of the CD4^+^ population into CD4^+^CD25^+^ and CD4^+^CD25^+^FoxP3^+^ in PMBC; this finding is in line with the significantly lower contribution of CD4^+^ compared to the i-p group. This result confirms the effective induction of the mucosal immune system response, constituting a natural pathway for emerging food allergens in the gut (Figure 3C,D).

### 3.4. Ex Vivo Lymphocyte Responses to α-la and β-lg

We conducted an ex vivo study to see how modified antigens affect T-lymphocyte differentiation. We stimulated isolated lymphocytes from the MLNs (Figure 4) and splenocytes (Figure 5) of the i-g- and i-p-sensitized groups with antigens before and after glycation.

The results showed that the MLN-derived lymphocytes of the i-g group stimulated with glycated β-lg/G increased CD4^+^ induction in a manner similar to that of ConA, which served as an antigen-independent positive control. The CD4^+^ T cells achieved a significantly higher percentage compared to the control and the other stimulator treatments, namely, those corresponding to the α-la, α-la/G/L, β-lg, and β-lg/L protein variants (*p* < 0.05), yielding a value of 30.15 ± 0.92% vs. 22.3 ± 0.001%, 20.85 ± 1.48%, 21.25 ± 0.35%, and 19.50 ± 0.001%, respectively (Figure 4A). None of the stimulated antigens increased the percentage of CD4^+^ cells derived from the i-p/β-lg group (Figure 4A). The primary antigen β-lg and that glycated with glucose and lactose and α-la glycated with lactose caused the ex vivo induction of CD4^+^CD25^+^ T cells from i-g/β-lg mice (Figure 4B). The cultured i-p group MLN lymphocytes, after stimulation with α-la and α-la/L, showed increasing CD4^+^CD25^+^ T cell percentages when inhibited by β-lg stimulation (Figure 4B) compared to the control. An almost two-fold-higher percentage of CD4^+^CD25^+^ T cells was found in the MLN lymphocyte culture from mice after oral β-lg exposure. At the same time, we observed a significant (*p* < 0.05) increase in the CD4^+^CD25^+^FoxP3^+^ population (Treg) when exposed to glycated α-la/L and β-lg/G (Figure 4B) compared to the control (29.6 ± 1.97%, 27.75 ± 0.35%, and 26.3 ± 0.42%, respectively). Lymphocytes from i-p/β-lg MLNs responded to stimulation with α-la and α-la/L and significantly induced CD4^+^CD25^+^FoxP3^+^ cells compared to the control (*p* < 0.05) (Figure 4B). The results suggest that oral β-lg exposure effectively activated the CD4^+^, CD4^+^CD25^+^, and CD4^+^CD25^+^FoxP3^+^ T cells involved in the modulation of the immune response to an allergen. Interestingly, double-positive CD4^+^CD8^+^ regulatory T cells (DPTs) were only found in the i-p/β-lg group (Figure 4D). The CD4^+^ helper and CD8^+^ cytotoxic T cells, two major subsets of lymphocytes, are differentiated from the common double-positive precursor CD4^+^CD8^+^ [31]. The absence of DPTs may indicate less severe inflammation in the i-g/β-lg group than that in the i-p/β-lg group. DPTs are considered a separate subpopulation of T-cells associated with different disease-specific functions. The relevance of DPTs in the pathogenesis of infections, tumors, and autoimmune disorders has been recognized [32]. The cytotoxic and/or immunosuppressive role of DPTs may indicate that these cells are heterogeneous and present pleiotropic functions that need to be investigated in further studies.

The spleen plays a pivotal role in regulating T and B cells’ responses to antigens circulating in the bloodstream. We cultured the experimental groups’ splenocytes, which were stimulated with native or modified 𝛼-la and 𝛽-lg (Figure 5). The cells from the i-g group were found to be more reactive to the stimulatory antigen than the cells from the i-p group. The percentage of CD4^+^ T cells increased significantly (*p* < 0.05) after 𝛼-la, 𝛼-la/G, or 𝛼-la/L stimulation, amounting to values of 14.09 ± 0.16%, 13.15 ± 0.21%, and 9.33 ± 0.01%, respectively, compared to that of 7.14 ± 0.01% in the control cells (Figure 5A). Stimulation with 𝛽-lg also significantly induced CD4+ T cells, presenting values of 13.10 ± 0.003%, 15.12 ± 1.25%, and 15.2 ± 0.85% after 𝛽-lg, 𝛽-lg/G, or 𝛽-lg/L stimulation, respectively (Figure 5A). The glycation of 𝛽-lg significantly increased the percentage of CD4^+^ cells compared to native 𝛽-lg stimulation and the control (*p* < 0.05). Splenocytes from the i-p group increased CD25^+^ induction almost five-fold after stimulation with 𝛼-la (10.70 ± 0.42%) or 𝛽-lg (11.70 ± 0.28%) compared to the control cells (2.94 ± 0.09%) (Figure 5B). Contact with glycated antigens decreased CD4 induction on the surface of T cells about two- to three-fold (*p* < 0.05) compared to the non-glycated form.

**Figure 5 nutrients-15-03110-f005:**
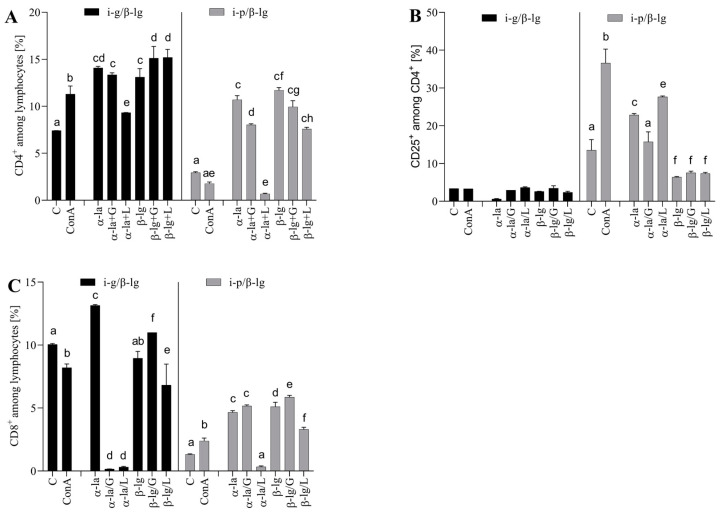
Splenocyte T lymphocyte profiles: (**A**) CD4^+^, (**B**) CD25^+^, and (**C**) CD8^+^ after stimulation with α-lactalbumin (α-la) and β-lactoglobulin (β-lg) and after glycation of antigens with glucose (α-la/G; β-lgG) or lactose (α-la/L), β-lg/L). Cells grown in medium only (C) and stimulated with Concanavalin A (10 μg/mL) (ConA) served as controls. Splenocytes were isolated from mice (*n* = 8) immunized intragastrically (black bars) and intraperitoneally (grey bars) immunized with β-lg. Results are presented as the means of the groups (n=8) ± SD. Different superscript letters denote statistically significant differences (*p* < 0.05).

Splenocytes from both experimental groups were cultured for a proliferation test. The representative charts in Appendix A show the splenocyte proliferation index (PI) after a variety of proteins’ stimulation. We found that oral antigen exposure (i-g) effectively induced CD4^+^ cell proliferation. PIs yielded values of 2.32, 2.51, and 2.15 for β-lg, β-lg/G, and β-lg/L stimulation, respectively (Appendix A). Lymphocytes from i-p/β-lg responded to stimulation in a weaker manner (Appendix A). We also found that glycation with glucose was more efficient for CD4^+^ T cell proliferation (Appendix A).

The activity of induced CD4^+^ T cells is crucial for achieving system-wide effects, leading to different immune reactions (inflammation, allergic reactions, or tolerance). Within the i-g MLN CD4^+^ subpopulation, native β-lg stimulation induced a 2-3-fold greater increase in CD4^+^CD25^+^ cells than that in the i-p β-lg group cells (*p* < 0.05) (Figure 4B). The opposite situation transpired during the stimulation of the i-p SPL lymphocytes with native α-la, where the percentage of CD4+ cells increased about eight-fold (Figure 5B). Litman et al. [30] demonstrated that CD4^+^CD25^+^ Treg supports immune tolerance and maintains intact immune responses after exposure to food allergens. Our observations are consistent with the trend observed by Litman’s group [30]. This suggests a basis for treating the i-g route of antigen administration as an effective form of tolerance induction. The presence of CD4^+^CD25^+^ Treg reflects an immune response to different environmental parameters, e.g., food allergens. The stimulation of i-g splenocytes with α-la, β-lg, and β-lg/L decreased the number of CD4^+^CD25^+^ cells. The resulting level was lower than the level observed in the control cells. β-lg glycation did not influence the percentage of CD4^+^CD25^+^ cells in the i-p β-lg group (Figure 5B). The glycation of α-la with lactose significantly decreased the differentiation of CD8^+^ lymphocytes in the splenocytes derived from the i-g and i-p groups. In the case of glycation with glucose, the CD8 percentage increased in the i-g group, while it decreased in the i-p group (Figure 5C). Overall, with regard to splenocyte culture, it is difficult to discern a clear pattern of the effect of glycation on the inhibition or stimulation of the analyzed cell population. This process seems to be dependent on the type of protein, the sugar used for glycation, and the pathway of immune response induction. Pitmon et al. [33] demonstrated the influence of different glucose concentrations on the percentage of Tregs. Ledesma-Osuna et al. [26] presented differences in the glycation process during the use of different saccharides. The study presented herein shows the effect of several variables (e.g., type of protein and sugar) on Treg induction. However, more detailed analyses are required to determine the mechanism behind the induction of regulatory cells via glycated antigens.

### 3.5. Cytokine Profiling

The regulation of immune responses by Th cells is accomplished through the secretion of specific cytokines. Th1 type response involves the secretion of pro-inflammatory cytokines (e.g., IFN-γ and TNF-α) and cell-mediated immunity, whereas Th2 differentiation is associated with food allergies and is mediated by interleukins (e.g., IL-10 and IL-17A) that activate mast cells and IgE-producing plasma cells, leading to humoral immunity. Table 1 shows the profiles of the cytokines released in the culture medium of the experimental groups’ splenocytes stimulated by α-la or β-lg before or after glycation.

We found that intraperitoneal immunization with β-lg induced more reactive T cells. The concentrations of IL-10, TNF-𝛼, IFN-γ, and IL-6 released in the media after stimulation with 𝛼-la and 𝛽-lg were significantly higher compared to those in the i-g splenocyte culture (*p* < 0.0001) (Table 1). The stimulation of cultures with glycated 𝛼-la and 𝛽-lg decreased or did not change the number of cytokines released. TNF-α initiated the inflammatory process. Its higher secretion by splenocytes from the i-p/𝛽-lg group highlights the difference in the pathway of the immune response induced by antigen administration. IFN-γ is associated with allergy-related immunopathologies. It is a major Th1 effector cytokine that reflects the development of inflammation. We found that i-p/β-lg splenocytes secreted significantly more of this cytokine than i-g/β-lg splenocytes. This casts intraperitoneal desensitization as an aggressive method of administering food allergens that does not reflect the natural pathway of the induction of the immune response by food allergens. IL-6 induces inflammation by increasing the synthesis of acute-phase proteins and neutrophils in the bone marrow. At the same time, it also aids in the suppression of inflammation by inhibiting TNF-α [34]. IL-6 promotes Th2 differentiation and simultaneously inhibits Th1 polarization [35].

IL-17A mediates crosstalk between the immune system and various epithelial tissues. This cytokine is released by Th17 cells; a subset of T cells and IL-6 are involved in this polarization [36,37]. The mechanisms of Th1/Th2 differentiation and their principles of action are well established. The pathway of Th17 cells in the development of food allergies is not fully understood. In the presented experiment, in all the samples analyzed, the IL-17A levels were higher for the i-g/β-lg splenocytes than they were for the i-p/β-lg splenocytes. The glycation of the protein molecule changed the protein’s immunogenicity, and the amount of secreted ex vivo IL-17 depended on the protein and the type of sugar.

Overexpression of Th2 can lead to inappropriate immune responses and, consequently, allergies and asthma. Overexpression of Th1 or Th17 can result in the development of autoimmune diseases such as rheumatoid arthritis and multiple sclerosis [38,39]. The dynamics of IL-17 release, the presence of the DP CD4^+^CD8^+^ population, and a significant percentage of CD4^+^CD25^+^FoxP3^+^ in the i-g/β-lg group suggested that inflammation developed upon initial contact with the milk allergen β-lg. Knowledge of the cytokines and T-cell populations involved in the Th17 pathway with respect to the development of the immune response to a food allergen may be crucial to establishing intragastric sensitization as a model for food allergy research.

## 4. Conclusions

This study confirmed that protein modification via glycation is a method for changing, but not decreasing, the immunogenicity of allergic proteins. The results demonstrated the importance of the sugar employed in the glycation process with respect to the effect on the immune system response. High doses of β-lg intragastric administration induced cellular immune responses that, in turn, induced Th1- and Th17-type cells, while the intraperitoneal route activated Th2 cells and induced significant inflammation and high IgE. The mesenteric lymph nodes are the inductive site of the immune response, and the i-p route of antigen delivery engenders an increased induction of CD4^+^ and CD4^+^FoxP3^+^ T cells in mesenteric lymph nodes. An ex vivo model with glycated whey proteins was used to prove that this modification did not decrease protein allergenicity. Stimulation with 𝛽-lg and 𝛼-la variants uncovered the important role of the Th17 pathway in the food allergen induction response, which could be crucial for allergy treatment.

The results of this study provide detailed information on how glycation changes bovine whey milk proteins during industrial/thermal processing and storage. This knowledge will be helpful in the development of milk-based products with characterized immunogenic potentials. Future studies are needed to determine trends in the standard food matrices and deeply characterize the pathway of the role of double-positive CD4CD8 T cells and the Th17 pathway in innate response modulation.

## Data Availability

The data presented in this study are available within the article and Appendix A.

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
