# Peer review of "Glycation of Whey Proteins Increases the Ex Vivo Immune Response of Lymphocytes Sensitized to β-Lactoglobulin"

_nutrients, 2023, doi:10.3390/nu15143110_

Round 1

Reviewer 1 Report

The manuscript explores the immunogenic properties of bovine milk proteins, namely α-lactalbumin (α-la) and β-lactoglobulin (β-lg), in their native and glycated forms. The authors examine these proteins' responses in vivo using C57BL/6 inbred mice and then conduct further in vitro tests on the isolated lymphocytes to identify immune reactions.

They employ a detailed glycation protocol and use SDS-PAGE to assess protein characteristics. The paper also outlines the protocols used for animal care and in vivo experimentation, including the methods for serum and fecal sample preparation, lymphocyte isolation, lymphocyte culture, and lymphocyte proliferation assays.

To summarize, the paper is an in-depth examination of the immune response to bovine milk proteins, with potential implications for understanding food allergies and immune responses to dietary proteins.

However, the manuscript requires major revisions to address concerns related to clarity, consistency, and structure.

Point 1: Ensure consistent use of terminology throughout the paper. For example, maintain consistency in the usage of terms like "centrifuged" and "centrifugation".

Point 2: Use more precise language to improve comprehension. For instance, instead of saying "for counting", it might be clearer to say "to count the number of cells".

Point 3: Make sure all abbreviations are defined at their first usage. It's essential to make clear what "CM" and "IM" stand for in lines 161-167.

Point 4: Improve structure and readability by breaking up longer paragraphs into smaller, more digestible chunks. For example, the paragraph starting at line 171 could be split into two or more smaller paragraphs.

Point 5: It appears that the discussion section of the paper is missing. The discussion section is critical as it helps interpret the results and links them to the wider context of research in the field. Make sure to include this section, detailing the implications of your results, how they support or contrast with other research, and possible directions for future study.

Author Response

Dear Editor:

Thank you for taking the time to review the manuscript. Thank you for all your comments. I truly believe that my corrections will satisfy your expectations and improve the quality of the manuscript.

Below is my response to each of the comments. In the text of the manuscript, all corrections are highlighted.

Point 1: Ensure consistent use of terminology throughout the paper. For example, maintain consistency in the usage of terms like "centrifuged" and "centrifugation."

The passive side of the verb is sometimes questioned during the language correction. Therefore, I sometimes used the active form of the verb to describe the action. Thank you for your attention. I tried to standardize the text.

Point 2: Use more precise language to improve comprehension. For instance, instead of saying "for counting," it might be clearer to say "to count the number of cells".

Thank you for the comment. I corrected it.

Point 3: Make sure all abbreviations are defined at their first usage. It's essential to make clear what "CM" and "IM" stand for in lines 161-167 (p.2.3.1).

Thank you for the comment. I added the abbreviation list at the end of the manuscript.

Point 4: Improve structure and readability by breaking up longer paragraphs into smaller, more digestible chunks. For example, the paragraph starting at line 171 could be split into two or more smaller paragraphs.

Thank you for the comment. I followed the manuscript, and I made some changes.

The methods part is built on two levels. The p.2.3 bind whole methods used in biological experiments. Point 2.3.1 describe sample collection and preparation; p.2.3.2 is a composition list of media used for the isolation of lymphocytes; p.2.3.3 describe PMBC isolation; p. 2.3.4 isolation of lymphocytes from tissues. Line 171 (version downloaded from the Editor) is the title of point 2.3.3. describing PMBC isolation only. I left it the same way; those few sentences described only this.

Point 5: It appears that the discussion section of the paper is missing. The discussion section is critical as it helps interpret the results and links them to the broader context of research in the field. Make sure to include this section, detailing the implications of your results, how they support or contrast with other research, and possible directions for future study.

The Autor guidelines allow for combining Results with Discussion. Thanks to your comment, I realized the section head lost the word Discussion. Anyway, I completed this part. The results are discussed with the bibliography positions starting from number 23.

Reviewer 2 Report

Short summary of article

Advance Glycation end product effect (Glu and Lac) on immune response intragastric and intra peritoneal of milk protein. 

Major comments

1)    Introduction  i suggest to write the reaction of the glycation products of protein  with monosaccharides

Write a model protein with amino groups, add a generic molecule of a reducing sugar (aldeyde group) and show the imine intermediate and the glycated product

2)    Line 48-49 do not confuse  AGEs (already reacted aldeydes) with AGEs precursors (not yet reacted aldeyde). Sugar (e.g glucose, lactose ... ) react only in one side and do not produce crosslinking in protein or tissue just make more stiky the surfice. Glyoxal, methylgyoxal present in all heat exposed food over 100°C. This molecules are crosslinker and produce glycated and crosslinked biomolecules.

3)    Line 69 add citation

4)    Line 121 NaIO4 write 4 in pedices NaIO4

Minor comment

            Line 59 : chitosan is a polyamine and it react with glucose and can be glycated by glucose more easly with respect to protein .

also ad a schematic picture of the glycated protein with glucose and lactose and show the result graphycally would for sure help the reader to undertand better the result 

Author Response

Dear Editor,

Thank you for taking the time to review the manuscript. Thank you for all your comments, which I hope will satisfactorily improve the quality of the manuscript.

Below is my response to each of the comments. In the text of the manuscript, all corrections are highlighted.

Major comments

1)    Introduction  i suggest to write the reaction of the glycation products of protein  with monosaccharides

Write a model protein with amino groups, add a generic molecule of a reducing sugar (aldeyde group) and show the imine intermediate and the glycated product

Response: Thank you very much for the suggestion. I prepared a scheme of the Maillard reaction and included it in the Introduction.

2)    Line 48-49 do not confuse  AGEs (already reacted aldeydes) with AGEs precursors (not yet reacted aldeyde). Sugar (e.g glucose, lactose ... ) react only in one side and do not produce crosslinking in protein or tissue just make more stiky the surfice. Glyoxal, methylgyoxal present in all heat exposed food over 100°C. This molecules are crosslinker and produce glycated and crosslinked biomolecules.

Response: Thank you for the comment, which I agree with. Raubach et al. (2020) experimented on myoglobin's glycation in glucose or methylglyoxal (MGO) presence. The authors checked the influence of myoglobin glycation on its proteasome degradation. They studied different glycation conditions - varying glucose concentration and one MGO concentration. Their results proved that "Severe glycation generated crosslinked proteins (…) Severe glycation again decreased proteolytic cleavage which might be due to cross-linking of protein monomers. (…) Protein crosslinking with MGO was performed." Proteasomal degradation is, as I understood, the physiological mechanisms of protein degradation. Their results showed that "Proteasomal degradation of modified myoglobin compared to native myoglobin depends on the degree of glycation: physiological conditions decreased proteasomal degradation whereas moderate glycation increased degradation. (…)" The Authors conclude that "chronic glycation under certain circumstances, e.g., hyperglycemia or aging, leads to extensive AGE formation and intermolecular crosslinks which the proteasome cannot degrade. Severe glycation might lead to AGE accumulation and potentially detrimental effects". and potentially detrimental effects."

3)    Line 69 add citation

Response: Thank you for this comment. I have completed the citation and rewritten the paragraph.

4)    Line 121 NaIO4 write 4 in pedices NaIO4

Response: Thank you, the chemical formula has been corrected.

Minor comment

            Line 59 : chitosan is a polyamine and it react with glucose and can be glycated by glucose more easly with respect to protein .

Response: I called chitosan a polysaccharide after the literature I cited. Sukhadeorao et al. (2019) define chitosan as "Amid, chitosan, the second most ubiquitous polymer after cellulose, exists as a β-(1–4)-linked d-glucosamine/N-acetyl-d-glucosamine randomly distributed linear polycationic yield from partial deacetylation of chitin polysaccharide. (…) As major polysaccharides are either neutral/negatively charged in an acidic environment, instead chitosan is cationic, eventually forms electrostatic complexes/multilayer structures/composites with anionic synthetic dopants/natural polymers."

Perhaps the structure and chemical groups in chitosan allow it to be classified in multiple ways. I am not arguing with that.

Nevertheless, thanks to your comment, the error has been discovered. Aoki et al. (2006) conjugated b-lg to chitosan, which probably changed the sentence's meaning. As a result, B-lg had lower immunogenicity. I corrected it.

also ad a schematic picture of the glycated protein with glucose and lactose and showing the result graphically would for sure help the reader to understand better the result 

Response: I added the scheme reaction to the Introduction as I wrote above. I hope you agree that the second time could be readable as duplicating.

Round 2

Reviewer 1 Report

Dear authors,

Your revised manuscript demonstrates a clear understanding of the experimental results and their implications in the field. The data analysis and interpretation provide valuable insights into the role of protein modification via glycation on the immunogenicity of allergic proteins and the specific immunological responses triggered by different routes of β-lg administration. Furthermore, the delineation of the role of Th17 pathway in food allergen-induced response is quite enlightening.

However, there are two key areas that could be further improved to enhance the completeness and applicability of your research findings.

- Firstly, the manuscript currently lacks a section detailing the limitations of your study. Discussing the limitations provides context for the interpretation of your results, identifies methodological constraints, and points out potential biases in the study design or execution. Including such a section enhances the transparency of your research process and could aid future efforts to replicate or build upon your work.

- Secondly, the manuscript would greatly benefit from an explicit mention of future research directions. While your discussion does implicitly suggest areas for future inquiry, explicitly outlining them can direct other researchers in the field and further underscore the significance and potential impact of your findings.

Including these points would strengthen your paper and provide readers with a more comprehensive understanding of your study.

Author Response

Dear Reviewer,

Thank you for your contribution to improving our manuscript. Below are our responses to the comments.

- Firstly, the manuscript currently lacks a section detailing the limitations of your study. Discussing the limitations provides context for the interpretation of your results, identifies methodological constraints, and points out potential biases in the study design or execution. Including such a section enhances the transparency of your research process and could aid future efforts to replicate or build upon your work

Thank you for this comment. Until now, We have been required to place this section only in project proposals. So, I set this paragraph at the beginning of the Results and Discussion section, following the guidelines in the article by Brutus, Stéphane, et al. Self-Reported Limitations and Future Directions in Scholarly Reports: Analysis and Recommendations. Journal of Management 39 (January 2013): 48-7. I hope it will find your acceptation.

I added the paragraph:

“The research presented here describes how whey protein glycation affects the cellular immune response. The strength of this research is the target, the glycation process of milk proteins. Glycation is a spontaneous process that accompanies the thermal processes of the food industry. Milk whey proteins (α-la and β-lg) are a nutrient-rich raw material commonly used in the food industry. Our study used a model system of single proteins, which limits the possibility of assessing the influence of interactions between proteins on the progress of glycation; the competitiveness of proteins in reaction with sugar. Glycation in the model system reduced the number of variables in the experimental setup that affect the induction of immunocompetent cells. This enabled us to determine the trend of changes in the immune system response at the cellular level.”

- Secondly, the manuscript would greatly benefit from an explicit mention of future research directions. While your discussion does implicitly suggest areas for future inquiry, explicitly outlining them can direct other researchers in the field and further underscore the significance and potential impact of your findings.

Thank you for this point. As a scientist, I hope you understand our reserve to present details of future research directions. In the conclusions, we have only given a general scope without indicating the specific recommendations we intend to take and publish in the future.

We add:

"Future studies are needed to see  this tendency in the standard food matrices and deeply characterize the pathway of the role of double positive CD4CD8 T cells and the part of Th17 pathway I innate response modulation."

Thank you one more time for your time directing to improving the manuscript.